# NDE Detection Techniques and Characterization of Aluminum Wires Embedded in Honeycomb Sandwich Composite Panels Using Terahertz Waves

**DOI:** 10.3390/ma12081264

**Published:** 2019-04-17

**Authors:** Kwang-Hee Im, Sun-Kyu Kim, Jong-An Jung, Young-Tae Cho, Yong-Deuck Woo, Chien-Ping Chiou

**Affiliations:** 1Department of Automotive Engineering, Woosuk University, Chonbuk 565-701, Korea; wooyongd@woosuk.ac.kr; 2Division of Mechanical System Engineering, Chonbuk Natl. University, Jeonbuk 561-756, Korea; sunkkim@chonbuk.ac.kr; 3Department of Mechanical and Automotive Engineering, Songwon University, Gwangju 502-210, Korea; jungja@songwon.ac.kr; 4Department of Manufacturing Design Engineering, Jeonju University, Jeonbuk 560-759, Korea; choyt@jj.ac.kr; 5Center for Nondestructive Evaluation, Iowa State University, Ames, IA 50011, USA; cchiou@iastate.edu

**Keywords:** T-ray, honeycomb sandwiches, aluminum wires, refractive index, fiber direction

## Abstract

For many years, scientists have been aware of the importance of terahertz waves (T-rays), which have now emerged as an NDE (nondestructive evaluation) technique for certain ranges of the electronic spectrum. The present study deals with T-ray scanning techniques of honeycomb sandwich composite panels with a carbon-fiber-reinforced plastic (CFRP) skin as well as the refractive index (**n**), and the electrical conductivity (**α**) of glass fiber-reinforced plastic (GFRP) composites. For this experiment, the degree of penetration to FRP composites is investigated for the THz transmitted power based on the angle in the electric field (E-field) direction vs. the direction of the unidirectional carbon fibers. Also, when CFRP skin honeycomb sandwich panels are manufactured for use in aerospace applications, aluminum wires are twisted together into the one-sided surface of the honeycomb sandwich panels to protect against thunderstorms. The aluminum wires are partly visible because they are embedded in the CFRP skin on the honeycomb sandwich panels. After finishing work with a paintjob, the wires become invisible. Thus, detecting the aluminum wires is a key issue for product monitoring. Based on a simple resistor model, an optimal scanning method is proposed to determine the preferred scan orientation on the baseline of the E-field in the direction of fibers to evaluate the level of transmission of T-rays according to the frequency bandwidth. Thus, the combination of angles required to detect the aluminum wires embedded with carbon fibers on the surface of the composite panels can be determined.

## 1. Introduction

The importance of terahertz waves (T-rays) in technical applications has been recognized in recent years, and T-ray detection fields within the range of the electronic spectrum have emerged due to the advancement of devices. The “T-ray gap” refers to the segment of the electromagnetic spectrum between microwave and infrared (0.1–10 THz) that remained largely unexplored until recent years due to the lack of a sufficiently bright source and a sufficiently sensitive detector. In particular, T-rays are critically important for security devices used in special airport regions, the medical area, polar liquids, various other industrial applications, and in the spectroscopic evaluations of composites [1,2,3]. In addition, terahertz time domain spectroscopy (THz-TDS) has taken an increasing role in determining the electrical field (E-field) of THz monitored transients via ultra-broadband electro-optic sampling and a simulation of the plane wave model [4].

A transit change in a THz emitter can be produced due to the resistance of photo-conductive switches over the THz timescale [5,6,7,8]. In addition, an optical anisotropic transformation or optical mixture can be used that can receive twin continuous wave (CW) lasers.

THz-TDS equipment is being used as a THz imaging device that is small, portable, and reliable enough for practical use since it not only has an unlimited range of applications but also includes excellent utilization of application technologies. The strength of fiber-reinforced plastics (FRP) has been observed in the space and civil aviation sectors in terms of T-ray applications, and T-rays have been applied to detect various foreign materials such as coatings, Teflon tapes, and debonding in FRP laminated plates [9]. Also, T-ray systems are able to detect shallow thicknesses like the ply thickness of FRP composites. Here, the wavelength in air at 1 THz is 0.3 mm, and the frequency range is 0.1 to 1.5 THz. In practice, nondestructive evaluation (NDE) techniques are widely established with an unlimited range of applications. T-rays can go through a vacuum and are not hampered by “shadow effects” (smaller cracks blocked by larger cracks); thus, T-ray NDE is emerging.

The present study deals with a method of measuring the refractive index (**n**) and the amount of T-ray transmitted power for both carbon-fiber-reinforced plastic (CFRP) and glass fiber-reinforced plastic (GFRP) composites and THz scanned images of CFRP composites based on their electrical conductivity. Carbon fibers are conductors whereas epoxy matrixes are not conductive; thus, the carbon fibers in CFRP laminated plates are conductive [10]. Also, when CFRP skin honeycomb sandwich panels are manufactured for use in aerospace applications, aluminum wires are twisted together into the one-sided surface of the honeycomb sandwich panels to protect against lightning and thunderstorms during flight [11]. After manufacturing, the aluminum wires are partly visible because they are embedded in the CFRP skin on the honeycomb sandwich panels, but after the work is finished with a paint job, the wires become invisible. Detection of these aluminum wires is an important for monitoring the correct wired locations when using these parts. This study therefore also focused on a method to detect the aluminum wires on the surface of conductive woven carbon fibers in CFRP skin honeycomb sandwich composite panels. 

Based on a simple resistor model, an optimal scanning method is proposed to determine the preferred scan orientation on the baseline of the electric field (E-field) in the direction of the fiber to evaluate the level of transmission of T-rays according to the frequency bandwidth. The combination of angles required in order to detect the aluminum wires imbedded with carbon fibers on the surface of sandwich composite panels can then be determined.

In this study, the results of the investigation on terahertz waves employed in nondestructive evaluations of FRP composites are summarized. Various experimental conditions for the fiber angle of samples and for the frequency based on the E-field were applied to examine the THz transmission power. A correlation was confirmed in the E-field vs. the direction of fiber in GFRP and CFRP composite laminated plates. Two kinds of CFRP and GFRP composites were tested using the THz experimental device. THz cannot transmit through CFRP composites because although carbon fiber is conductive, usually, CFRP composites are composed of carbon fiber and epoxy resin. The resin is nonconductive; so, the terahertz waves go through the carbon fiber composite laminates to some degree.

Assuming that the classical skin depth (δ) formula is valid for CFRP, we can then estimate the penetration depth of the terahertz waves into CFRP. For example, at 0.1 THz, δ ≈ 0.5 mm, and one ply of CFRP is about 0.125 mm, so the penetration is less than two plies at 1 THz and about four plies at 0.1 THz. These values seem to be reasonable in view of the TDS images of the flaws in the CFRP sample. A detection technique for aluminum wires in the composite panels is proposed based on a function of angles in order to determine the preferred scan orientation.

## 2. Basic Theory Approach 

The method presented here introduces a refractive index through the sample in the through-transmission mode in the T-ray time domain. Figure 1 shows the direction of the THz wave signals. When the T-ray passed through samples in the THz pulsed emitter, a refractive index was obtained by calculating a time-of-flight (TOF) while passing through the sample of arbitrary thickness and the travel time to the pulsed receiver [2].

### 2.1. Measurement of the Through-Transmission Mode

A T-ray TOF without a sample is calculated as follows:(1)$$T_{total\;time\;without\;sample}=\frac{L}{V_{air}}$$
Here, $T_{total\;time\;without\;sample}$ refers to the travel time to the receiver after the T-ray is generated at the emitter, and L refers to the distance between the emitter and receiver of the T-ray. $V_{air}$ refers to the speed of light ($3\times 10^{10}\mathrm{cm}/s$).
(2)$$n=1+\frac{\Delta_{t}V_{air}}{t}$$
Here, *n* refers to a refractive index (n) of the sample and d refers to the thickness of the sample.

Next, an absorption coefficient (α) of the sample in the through-transmission mode was determined using a generally accepted method [3].

First, the T-ray transmission time ($\Delta_{t}$) and the amplitudes of two samples, which were the same type but with different thicknesses, were calculated. The difference in the transmitted E-fields between the thicker and thinner samples was calculated via the following equation to induce an absorption coefficient (α):(3)$$\alpha =\frac{\ln\frac{I_{2}}{I_{1}}}{t_{1}-t_{2}}$$
where $I_{1}$ and $I_{2}$ refer to the transmitted E-fields of the thin and thick samples, $t_{1}$ and $t_{2}$ refer to the actual thicknesses of the thinner and thicker samples, and α denotes an absorption coefficient.

### 2.2. Measurement of the Reflection Mode

The index of refraction was solved by the below method. The different time in the optical path length could be determined between the plane and back wall reflection signals in the time line. A scheme displaying the system of the dual T-ray waves is shown as in [2]:(4)$$∴n^{4}-{\mathrm{An}}^{2}-{\mathrm{Asin}}^{2}\theta_{p1}=0$$
$$A=\left(T^{2}V_{\mathrm{air}}^{2}\right)/\left(4d_{2}^{2}\right)$$
where T is the transmitting time of the specimen, d_2_ is the specimen thickness, V_air_ is the velocity of light, and θ_p1_ is the angles of incident angle in the specimens. 

## 3. Experimental System and Measurements

### 3.1. Measurement System

A THz time-domain spectroscopy (THz-TDS) system, which is a nondestructive testing device, was utilized. The material characteristics and scanned images of samples were obtained using this system. The THz system used in the present study was manufactured by TeraView (UK). The system consisted of a time-domain spectroscope (TDS) pulse device and a continuous wave (CW) device. It consisted of TDS technology that generated, adjusted, or detected THz pulses. The T-ray beam was concentrated on focal distances of 50–150 mm, and full widths at half-maximum were designed with dimensions of 0.8–2.5 mm, respectively. The TDS device could be set up for the measurement in reflection and through-transmission mode. The frequency range in the CW device was 50 GHz–1.5 THz. The focal distances of the CW device were also 50–150 mm. The THz-TDS and THz-CW devices were connected via optic fibers. The THz-TDS system had its own E-field. Both systems were capable of performing high-speed imaging up to a speed of 5 cm/s at a mechanical resolution of 15 microns. The wavelength in air at 1 THz is 0.3 mm. The angles between the optical heads were 16.6 and 5.1 degrees for 50 mm and 150 mm foci, respectively. The step size was 0.5 mm for scanning, the fast delay line was up to 300 ps, and the time range was up to 1300 ps. The beam diameters were 0.7–0.9 mm (50 mm) and 2.25–2.65 mm (150 mm).

### 3.2. Measurement Method

Figure 2 shows the THz measurement system schematic in the reflection mode. During the experiments, THz waves were generated at the emitter and then sent to the receiver. Here, to conduct the experiment, the sample was positioned at a place where the focal points of the emitter and receiver were matched. The focal distances of the THz lens were 50 mm and 150 mm.

## 4. Results and Discussion

### 4.1. Measurement of the Terahertz Refractive Index

In order to measure T-ray parameters representing the physical properties of materials, THz pulses were sent in the reflection mode for poly methyl methacrylate (PMMA) materials. Figure 3 clearly shows an A-scan image of the reflection mode at the surface of the sample taken using the TDS system. The PMMA sample was prepared at a thickness of approximately 6.1 mm. The time difference (Δt), measured at the surface of the sample, was 63.5 ps, allowing the optical time difference to be calculated using the mode of reflection—one of the measurement techniques used in order to compute a refractive index. As shown in Table 1, samples of PMMA, GFRP composites, and fused quartz were measured using the reflection mode. When compared to data from previous studies, only a 1–2% difference was noted [2,3,10].

During the experiments, a variety of parameters relating to the refractive index were considered for measurement during the reflection and transmission modes. We note that since the GFRP composite samples were different from existing samples in terms of manufacturing method and characteristics, it was difficult to perform a comparison with existing data.

### 4.2. Evaluation of the E-Fields of Glass and Carbon Fibers 

T-rays can easily penetrate dielectric materials in contrast with conductive materials. Although THz was applied in the investigation of carbon fiber composites, it was necessary to produce more in-depth research results. CFRP composite laminates consist of conductive carbon fiber and non-conductive resin. When CFRP composite laminates are observed via microscope, they consist of two main components—carbon fiber and resin—allowing significant conductivity of fibers. A quantitative evaluation of the T-ray characteristics of carbon fiber composites is needed due to these findings. A number of publications [12,13] have reported that the value of conductivity ($\sigma_{l}$) in the longitudinal direction was in the range of 1 × 10^4^ to 6 × 10^4^ S/m. The data range for the transverse conductivity (σ_t_) is particularly wide ranging from 2 to 600 S/m [12]. 

The conductivity value in the transverse direction of the laminated plate using prepreg sheets, which are a unidirectional oriented composite, varied significantly according to the quality of the laminated plate and the manufacturing process. The plane conductivity with regard to the current that flows with angle θ in the fiber axis in unidirectionally oriented CFRP composite can be calculated as follows [13]: (5)$$\sigma =\sigma_{l}cos^{2}\theta +\sigma_{t}cos^{2}\theta$$

Since much greater conductivity in the fiber length direction ($\sigma_{l}\gg \sigma_{t}$) was shown, the THz wave that was transmitted in the unidirectionally oriented CFRP composites was significantly different according to the relative angle between the carbon fiber axis and electric field vector. When the E-field of the THz was parallel to the carbon fiber axis direction, conductivity became highest while transmitted power became the lowest. When the direction of the E-field was perpendicular to the angle of the fiber, conductivity became the lowest and transmitted power became the highest.

The transmitted power at the lower section (f–0.1 THz) of the frequency spectrum was higher than that of the noise level by more than 30 dB for the THz system. The transmitted power depends on the functions of angles with the frequency of THz waves for both GFRP and CFRP composites shown in Figure 4 using the CW photo-mixing system. For the THz testing, the through-transmission mode was used with a 50 mm focus lens. The frequency range was 0.1–1.5 THz. Here, the average of 100 data values was acquired. We note that a.u. refers to arbitrary units. 

Figure 4a shows the degree of transmitted power using 24-ply GRFP composites and Figure 4b shows the result using a 24-ply CFRP composite. Figure 4a does not exhibit much difference between the functions of angles and the transmitted power, even with changes in frequency and Figure 4b shows a big difference in the degree of transmitted power based on the function of angles. It was confirmed that the value of THz transmitted power depended on the angle between the direction of the E-field and the direction of the unidirectional carbon fibers. In particular, an optimal preferred scan orientation at a 90° scanning angle is suggested.

### 4.3. THz Imaging of Al Wires in Conductive CRFP Laminated Plates

In our analyzed approach, through extensive experimentation similarities and contrasts were found for both T-ray and ultrasonic testing (UT). Both follow Snell’s law in regard to refraction (“waves are waves”); velocity and attenuation are key quantities for both; strong similarities can be seen in the pulsed signals; and both can have A-, B-, and C-scans. However, T-rays can go through a vacuum, while UT cannot; UT can penetrate most solids, while T-rays can only penetrate non-conducting materials; UT is hampered by the “shadow effect” (a smaller crack blocked by a larger crack), while T-ray is not; and UT is a mature technology, while T-ray NDE is an emerging technology. 

Thus, we can assume that the classical skin depth formula is valid for CFRP conventional models given the above physical phenomena. For applications, we can then estimate the penetration depth of the terahertz waves in CFRP using the classical skin depth formula. We calculated the skin depth for a unidirectional CFRP with fibers normal to the E-field of the waves. We can estimate the skin depth using Equation (5):(6)$$\delta =\sqrt{\frac{2}{\omega \mu \sigma }}=\sqrt{\frac{\rho }{\pi f\mu }}$$
where ρ is the resistivity, ω is the angular frequency, μ is the permeability in all the conductors, and σ is the conductivity. Here, the skin depth is approximately 0.5 mm (or about four plies) at 0.1 THz normal to the fibers and σ_t_ = 10 s/m.

Therefore, because the effect of conductivity could be evaluated in the terahertz wave experimentations, the composite panels were used with CFRP skins. The aluminum wires were twisted together with the surface woven ply of the sandwich composite panels. The wires were closely woven within the CFRP skin at the surface. Aluminum wire is used in this manner in advanced materials to absorb lightning during flight. Nondestructive detection is very important issue for monitoring the exit of aluminum wires since it is a critical part design factor in airplanes.

Assuming that an E-field is vertical in the fiber length direction and horizontal in the fiber radial direction, the difference between the two angles is called θ. Here, foreign material was detected by using the reflection mode of TDS-THz waves, thereby obtaining T-ray images. T-ray penetration is highly dependent on the conductivity of the fibers and fiber orientation with respect to the T-ray electric field. The lowest conductivity is at 90 degrees, when the E-field is perpendicular to the fiber. The simple “resistor” model treats each ply as a resistor and all plies are connected in a parallel “circuit” so that the preferred scan orientation can be obtained. Thus, the simple resistor model can provide guidance in determining the preferred scan orientation by using the equation below. The conductivity is calculated from the passing current forming angle θ with the carbon fiber in CFRPs [13].
(7)$$\sigma =\sigma_{l}cos^{2}\theta +\sigma_{t}sin^{2}\theta$$

The correlation between conductivity and the S/N ratio of T-ray defect images is expressed as $\sigma_{l}\gg \sigma_{t}$, so Equation (5) can be expressed as $\sigma ≅\sigma_{l}cos^{2}\theta$. Here, conductivity was minimized when an E-field was generated at a 90° angle with a single ply using the equation $\sigma ≅\sigma_{t}sin^{2}\theta$. The signal-to-noise (S/N) ratio of the image was largest when a sample was placed at this angle. As such, the S/N ratio was the worst at the angle of a specimen that had the largest conductivity, which was $\theta =0^{{}^{\circ}}\;\left(\sigma =1.0\sigma_{t}\right)$. This tendency of the wire detection resolution indicated that an S/N ratio based on conductivity qualitatively matched experimental results [2,3]. As shown in Equation (5), the modeling conductivity of a woven ply was considered, assuming that they used two like resistors. $\sigma_{t}$ was ignored due to its very low values and R_eq_ = ($\sigma_{l}$)^−1^ was utilized in a correction with the S/N of the T-ray C-scan image. Here, R_eq_ was considered to be the resistance. The woven ply consisted of both 0° and 90° orientations. The two conductivities $\sigma_{1}\;\mathrm{and}\;\sigma_{2}$ thus had to be considered, as shown in Table 2.

Figure 5a shows a photo of the aluminum wires on the surface of the composite panels. Here the wires were twisted together via a pre-set process. Figure 5b shows a cross-sectional view of the panels. Figure 5c shows the end skin structure consisting of a woven aluminum wire followed by two plies of woven CFRP on the surface of the specimen. The end skin was bonded with the epoxy on the honeycomb cells. Figure 5d shows the T-ray testing configuration on the baseline of E-field direction according to the functions of angels in order to detect the aluminum.

Figure 6 shows 2-D scan images (*x* and *y* axes) of the TDS reflection mode, which are plotted based on the peak-to-peak amplitude. Here the plane of the scanned image is the *x*- and *y*-axis and the direction of thickness is the *z*-axis. The images were taken at every 22.5° to find the preferred scan orientation of the honeycomb sandwich composite panels. Here, the lower-resolution images are considered to have been affected by the rough surface of the sample. Firstly, Figure 6a shows the THz scanned image of an aluminum wire in the 0° direction of the CFRP skin in a CFRP skin sandwich with composite panels utilizing the TDS system. 

The detection of aluminum wires seemed to be very weak because the aluminum wires were twisted together into the surface ply of the honeycomb sandwich composite panels. The direction of the skin carbon fiber is perpendicular to the angle of E-field direction of the THz waves, which formed with the angles of 0°/90°. Here, the woven CFRP skin formed with an angle of 0° significantly affected the image. Figure 6b displays the C-scan result when the angle of the first CFRP skin locations was 22.5°. This image shows the two weak aluminum wires in the angle of 22.5°. Figure 6c displays the scanned result where the angle of the first CFRP skin was 45°. This image clearly shows the aluminum wires in the angle of 45°. The S/N of the AL wires was expected to be the highest at this angle to acquire scanned images of 0°/90° aluminum wires. Here, Point A indicates aluminum wires imbedded in 0/90 woven fibers, which are shown by dotted lines. 

Figure 6d shows the scanned image when the angle of the first CFRP skin was 67.5°. This image shows two weak aluminum wires in the angle of 22.5°. Figure 6e displays the THz C-scanned result at the angle of 0°, in which the aluminum wire was very weakly revealed. Here, in the case of θ = 0°, poor S/N ratios made for very weak images. Particularly, since the surface CFRP skin was two-ply, a comparison of the resistance in both plies was made in one image. Since THz waves cannot penetrate in the case of angles of both 0° and 90°, the image was very weak. The aluminum wires were revealed most clearly at the angle of 45°. This was due to the resistance of one ply significantly affecting the respective results at the angles of 22.5° and 67.5°. Thus, angle combination provides an optimal scanning method for determining the preferred scan orientation based on the simple resistor model.

Detection of the degree of defect varies according to the function of angles between the E-field and the direction of fiber. Table 2 shows the conductivity of the woven CFRP skin of composite panels using the modeling equation. Here, θ refers to an angle made between the first ply fiber axis and the electric field vector, and ψ refers to an angle made between the second ply of the fiber and the electric field vector. It shows the following cases: θ = 0°, θ = 22.5°, θ = 45.0°, θ = 67.5° and θ = 90.0°.

The first ply of the woven CFRP skin had minimum conductivity when the relative angle was 90° between the direction of the fibers and the direction of the E-field. As shown in Figure 6a, two *x*-axis wires in the T-ray image were observed most easily, but one *y*-axis wire in the T-ray image seemed to be unclear. Figure 6c shows a clear image for both of the two *x*-axis wires and the one *y*-axis wire on the surface of the composite panels. Note that Point A indicates a typical AL wire imbedded on the honeycomb composite panel as shown in Figure 6c. One ply of woven CFRP skin was needed to consider two-directional characteristics because the fibers were stacked with a [0/90] configuration. Each conductivity was considered for the two directions in each ply of woven CFRP skin due to the [0/90] stacking sequence of the woven CFRP skin. As shown in Table 2, the amount of modeled conductivity seemed to be the same, whereas each T-ray image had a different resolution. The two *x*-axis wires of Figure 6b and Figure 6d,e were not observed clearly because carbon fibers blocked T-ray penetration. In particular, THz waves could not easily penetrate the CFRP skin and detect the aluminum wires if the amount of conductivity became the maximum in all the other layers, as shown in Table 2. In addition, Figure 7 shows the relation between the normalized resistance and the E-field according to the fiber orientation of a 24-ply unidirectional CFRP composite laminate. In particular, when the E-field was 90° in the fiber orientation direction of a 24-ply unidirectional CFRP composite laminate, the resistance was the lowest, allowing a detected signal to be minimized as the transmission power of the THz wave was very high. 

Thus, when the CFRP skin was twisted together with aluminum wires into the surface fibers of the honeycomb sandwich composite panels, the aluminum wires in the images could be observed most clearly at the angle of 45°; however, the wires could not be clearly seen at both 0° and 90° because the woven CFRP skin partly blocked T-ray penetration when the woven CFRP skin was stacked at 0/90°.

## 5. Conclusions

The application and utilization of terahertz (THz) waves was investigated in the non-destructive testing of honeycomb sandwich composite panels, and a measurement technique for the refractive index of THz waves, as one of the material physical properties, was established. Although it is difficult for THz waves to be transmitted through carbon fiber composites, the degree of transmitted power of THz waves was compared with that through GFRP composites according to the frequency bandwidth. In addition, the test validity of THz waves according to carbon fiber orientation was investigated when aluminum wires were twisted together into the surface fibers of honeycomb sandwich composite panels. As a result, the following conclusions were made: 

(1) It was found that the refractive indices in the reflection and transmission modes could be measured using THz waves, and the degree of THz transmitted power depended on the angles between the E-field direction and the direction of unidirectional carbon fibers, of which the anisotropic electrical conductivity interacted strongly with E-field bases. 

(2) It was determined that the conductivity values of woven CFRP skin on the honeycomb sandwich composite panels could be obtained based on the simple resistor model, and they corresponded to the resolution of T-ray images. The images of AL wires were mapped out by applying optimal THz waves, which provides guidance in determining the preferred scan orientation.

(3) Since the THz waves are influenced by the combination angle of 0°/90° fibers in the CFRP skin on the honeycomb sandwich composite panels, the aluminum wires can be optimally detected at the angle of 45° as determined from the correlation with the E-field and the highest S/N ratio. 

## Figures and Tables

**Figure 1 materials-12-01264-f001:**
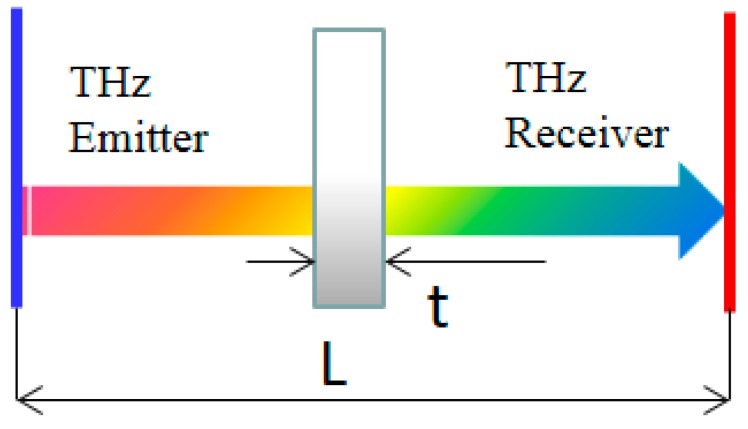
Schematic setup of terahertz wave (T-ray) testing in the through-transmission mode.

**Figure 2 materials-12-01264-f002:**
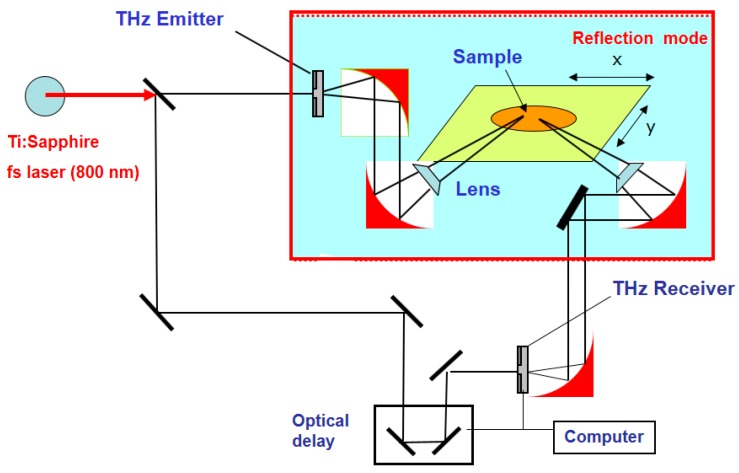
A simplified block scheme of the THz measurement method.

**Figure 3 materials-12-01264-f003:**
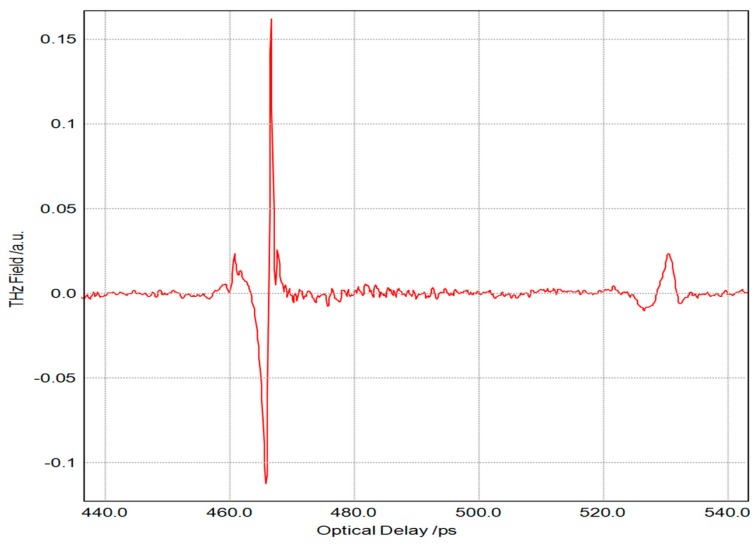
THz time-domain spectroscopy (TDS) pulses from a transmitted PMMA specimen.

**Figure 4 materials-12-01264-f004:**
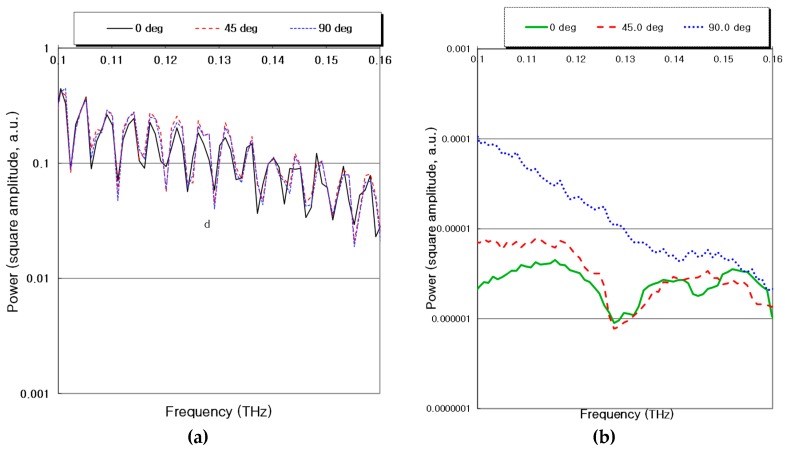
Relations between frequency of THz terahertz waves and transmitted power in continuous wave (CW) photo-mixing signals for a 24-ply glass and carbon composite laminates with the different functions of angles. (**a**) Glass fiber-reinforced plastic (GFRP) composites; (**b**) Carbon-fiber-reinforced plastic (CFRP) composites.

**Figure 5 materials-12-01264-f005:**
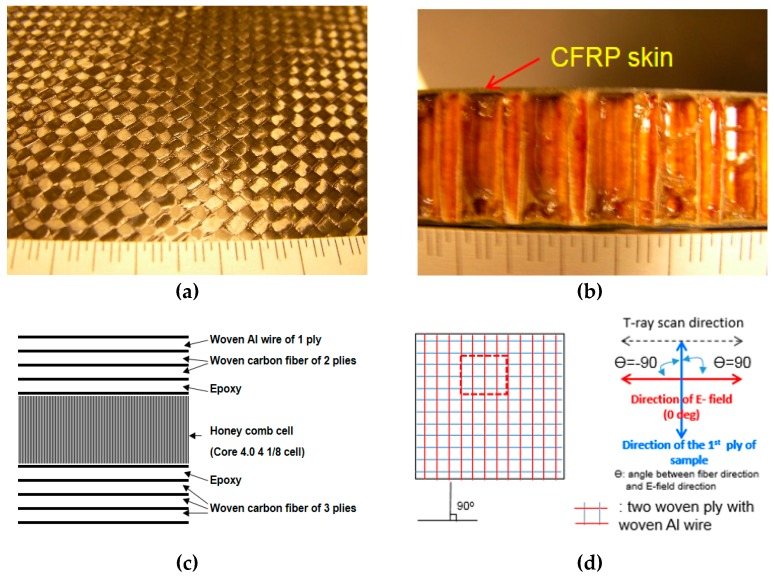
Overall diagrams for the terahertz reflection mode and CFRP skin sandwich composite panels with Al wires embedded on the surface. (**a**) Al wires on a sandwich panel; (**b**) Cross section of a sandwich panel; (**c**) Configuration of cross section; (**d**) T-ray testing configuration.

**Figure 6 materials-12-01264-f006:**
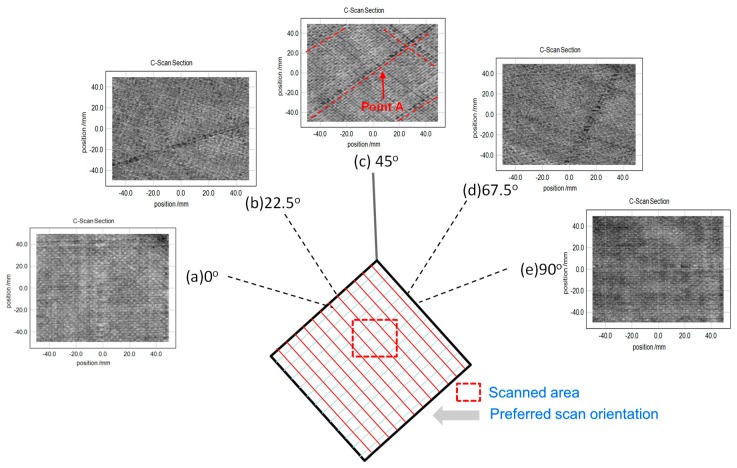
Terahertz scan images of the TDS reflection mode on embedded Al wires in honeycomb sandwich composite panels. (**a**) 0° scan image; (**b**) 22.5° scan image; (**c**) 45.0° scan image; (**d**) 67.5° scan image; (**e**) 90.0° scan image.

**Figure 7 materials-12-01264-f007:**
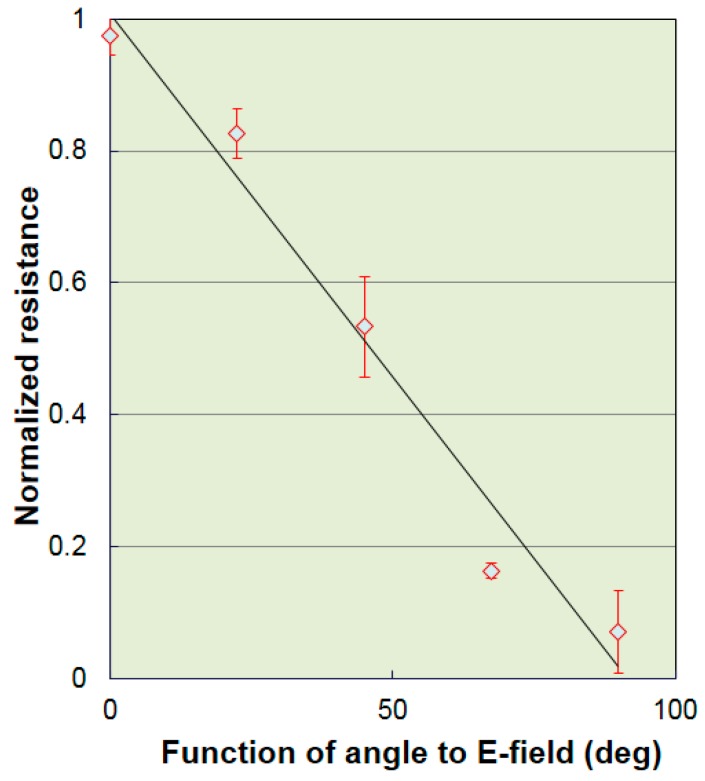
Relations of the function of E-field angle and normalized resistance with fiber angles in 24-ply CFRP laminates.

**Table 1 materials-12-01264-t001:** Averaged data on the T-ray “n” values of the materials researched.

Materials	Refractive Index (n) *	Refractive Index (n)
		Reflection Mode
PMMA	1.60 ± 0.08	1.58 ± 0.07
Fused quartz	1.95 ± 0.05	1.95 ± 0.15
GFRP	-	2.17 ± 0.15

Note: * Data in References [2,3,10].

**Table 2 materials-12-01264-t002:** Predicted conductivity of CFRP ply using on the simple resistor model.

Resistance	Angles				
	0°	22.5°	45.0°	67.5°	90.0°
θ	0°	22.5°	45.0°	67.5°	90.0°
ψ	90°	67.5°	45.0°	22.5°	0.0°
σ1	1.0	0.85	0.5	0.15	0.0
σ2	0	0.15	0.5	0.85	1.0
σ	1.0	1.0	1.0	1.0	1.0
Req	1	1	1	1	1

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
