# Peer review of "NDE Detection Techniques and Characterization of Aluminum Wires Embedded in Honeycomb Sandwich Composite Panels Using Terahertz Waves"

_materials, 2019, doi:10.3390/ma12081264_

Round 1
Reviewer 1 Report
Dear Authors,
I carried out this review third time and I still think that detection of the wires by THz waves could be of interest for readers of the Journal. Moreover, I do appreciate the effort you made for improvement of the manuscript, but my recommendation is still “Reconsider after major revision”.
Unfortunately, I found out that the manuscript still requires a lot of work to be understood.
1. There are many strange sentences which should to be rewritten. It is very hard to focus on Your research, if you must think hard what Authors meant. It strongly influences my assessment of this work.
2. In section 4 you completely mixed description of two THz techniques - TDS and CW photomixing. I don’t know which experimental results were obtained with which technique.
3. Quality of Figure 6 is still too low and some my previous recommendations were not meet, e.g. “Can you add description of z-axis?”
4. In previous review I suggested “Fig. 6 – quality is unacceptable. Can you mark important features? Can you add description of z-axis? How the c-scans were prepared? Maximum of amplitude of the signal?”
And in your explanation you gave some answers, e.g. “Scan image was plotted based on the peak-to-peak amplitude.”, which is OK, but I cannot find this explanation in the manuscript.
5. There is figure 7 caption but there is lack of figure 7 in the manuscript. !!!
In my opinion you should carefully rewrite the whole manuscript and improve quality of figures. Extensive editing of English language and style is also required.
Only then the manuscript can be scientifically reviewed.
Author Response
Dear First reviewer;
We do appreciate your very kind comments you made again. We will do our best for your questions and comments. Also, we are sorry for not enough delivering right answers you asked out all the time.
Please find one file attached herewith.
Thank you..

Reviewer 2 Report
Reviewer's report on "NDE Detection Techniques and Characterization of Aluminum Wires Embedded with Honeycomb Sandwich Composite Panels Using Terahertz Waves", by Kwang-Hee Im et al.
The authors employ high-frequency electromagnetic waves in the terahertz regime for nondestructive evaluation of fiber-reinforced plastic composites. The paper is an interesting piece of original work and deserves publication. However, I think the authors should revise their paper to sufficiently motivate their work and make the paper easier to understand for prospective readers. In particular, I have the following points:
(i) The authors should motivate the use of electromagnetic waves in the THz regime. From the paper it is unclear why a frequency of about 100 GHz should be used to test their FRP samples. Could not the same measurements be done with lower frequencies (e.g., with a microwave microscope at 4 GHz or so ?). This would reduce the experimental effort considerably.
(ii) The authors use the conventional skin depth to calculate the penetration of THz waves into their samples. I wonder if at these frequencies this model is valid. Although the resin matrix into which the carbon fibers are embedded is nonconducting at low frequencies, there is a capacitive coupling between the carbon fibers causing an rf current to flow even in the resin. This might reduce the penetration depth drastically. The skin depth finally used (0.5 mm at 100 GHz) seems to me to be too high; this would require nearly nonconducting carbon fibers. I made numerous measurements myself on CFRP samples and found such penetration depths at much lower frequencies (10 to 100 MHz).
(iii) The authors found that the signal-to-noise ratio for CFRP samples in the best case was 30 dB (line 192). The SNR is dependent on the bandwidth and measurement (averaging) time. Can the authors give data on how long a measurement took in order to achieve this SNR. This might give a prospective reader of the paper an estimate how practical the method might be. It would also be interesting to comment on the best-case SNR of a defect in the sample, and how small a defect could be detected with a certain SNR.
(iv) On page ten, the authors come to four conclusions. I think the first two of these could be omitted, as these are well known already. I wonder if the third conclusion 'conductivity values of woven CFRP skin on the honeycomb sandwich composite panels could be obtained based on the simple “resistor” model and were corresponding to the resolution of T-ray images' is really correct. Surely, not only the skin depth will determine the resolution of the images, but also the wavelength and focussing of the THz waves, so the skin depth could in reality be much lower.
(v) The authors should critically check the English language of the paper again. Sometimes there are confusing or misleading statements, which -when better expressed- could be easier understood. For example, in the abstract '... the CFRP skin is partly embedded with conductivity through which the level of transmitted power of T-ray was compared with that of glass'. A power cannot be compared through conductivity. On line 172, 'a value of conductivity in the lateral direction was much wider and ...' Does this mean that the spread in conductivity was larger than in the other direction ? On line 189 'When the direction of the E-field was parallel to the angle of the fiber ...', does this mean that the E-field was perpendicular to the fiber ?
(vi) Some minor points: The nice 2D scans shown in Fig. 6 seem to have too low a resolution. Even when zooming in, details in the scans are hardly visible. I wonder if the authors could provide larger images of the scans; probably not all scans at different angles are necessary. Fig. 7 was somehow missing from my review copy, so I cannot comment on it.
Author Response
Dear Second reviewer;
Thank you very much again for reviewing our paper on kind and valuable comments. We will do our best for your questions and comments. The answers about your comments are provided in this reply.
Please find one file attached herewith.
Thank you...

Round 2
Reviewer 1 Report
Dear Authors,
I think that you have met majority of my recommendations. Thank you.
My recommendation is “Accept after minor revision”.
Can you please improve quality of Fig.6 and add description of z-axis?
Fig. 7 – should be “normalized resistance”.
Author Response
We do appreciate your very kind comments you made again.
Anyway the answers about your comments are provided in this reply.
